# Juvenile Idiopathic Arthritis: A Review of Novel Diagnostic and Monitoring Technologies

**DOI:** 10.3390/healthcare9121683

**Published:** 2021-12-04

**Authors:** Amelia J. Garner, Reza Saatchi, Oliver Ward, Daniel P. Hawley

**Affiliations:** 1The Medical School, University of Sheffield, Sheffield S10 2TN, UK; 2Industry and Innovation Research Institute, Sheffield Hallam University, Sheffield S1 1WB, UK; r.saatchi@shu.ac.uk; 3Department of Paediatric Rheumatology, Sheffield Children’s Hospital, Sheffield S10 2TH, UK; oliver.ward1@nhs.net (O.W.); daniel.hawley@nhs.net (D.P.H.)

**Keywords:** juvenile idiopathic arthritis, rheumatoid arthritis, artificial intelligence, thermal imaging, accelerometry, fuzzy logic

## Abstract

Juvenile idiopathic arthritis (JIA) is the most common rheumatic disease of childhood and is characterized by an often insidious onset and a chronic relapsing–remitting course, once diagnosed. With successive flares of joint inflammation, joint damage accrues, often associated with pain and functional disability. The progressive nature and potential for chronic damage and disability caused by JIA emphasizes the critical need for a prompt and accurate diagnosis. This article provides a review of recent studies related to diagnosis, monitoring and management of JIA and outlines recent novel tools and techniques (infrared thermal imaging, three-dimensional imaging, accelerometry, artificial neural networks and fuzzy logic) which have demonstrated potential value in assessment and monitoring of JIA. The emergence of novel techniques to assist clinicians’ assessments for diagnosis and monitoring of JIA has demonstrated promise; however, further research is required to confirm their clinical utility.

## 1. Introduction

Juvenile idiopathic arthritis (JIA) is defined by the International League of Associations for Rheumatology (ILAR) as arthritis of unknown aetiology, starting before the sixteenth birthday, lasting longer than six weeks, with all other diagnoses excluded [1]. Arthritis itself is defined as swelling or effusion, increased warmth and/or painful limited movement with or without tenderness [2,3].

The JIA diagnosis encompasses distinct sub-classifications, defined by the ILAR in 2001 to include systemic-onset arthritis, oligoarthritis, polyarthritis rheumatoid factor positive, polyarthritis rheumatoid factor negative, psoriatic arthritis, enthesitis-related arthritis and undifferentiated arthritis [1]. This classification system has faced criticism as there is increasing evidence that some of these categories are more heterogenous and may be more accurately defined by a new system [4]. In 2019, the Paediatric Rheumatology International Trial Organization (PRINTO) began the verification process for a new classification system [4].

JIA is the most common rheumatic disease of childhood, with an average prevalence of 70 in 100,000 in Europe [5]. The data on the prevalence of JIA are varied, depending on disease classification, geographical area and study design and have been reported to vary between 3.8 to 400 cases per 100,000 worldwide [2]. JIA more commonly affects females than males (>2:1); however, this distribution varies within the disease classification system [6]. The age of onset of JIA also varies significantly between different subtypes; for example, the median age of presentation of systemic-onset arthritis has been reported as 2 years [7], whereas for enthesitis-related arthritis, this has been reported as 11 years [6].

JIA may affect any joint and frequently causes extra-articular inflammation (e.g., uveitis). When children and young people are being assessed for arthritis, it is important that a thorough and careful examination is undertaken, as some joints, e.g., the temporomandibular joint (TMJ) and hip joint, are unlikely to demonstrate signs of swelling. The involvement of TMJ in JIA was recognized in 1897 [8]. Diagnostic imaging is particularly important in assessing certain joints such as TMJ [9]. Avoiding delay in diagnosis of arthritis in these joints is important, as joint damage can result in functional disability; for example, the functional implications of TMJ arthritis include difficulty in eating, pain with talking and alterations to the facial appearance [10]. Further reviews and discussions of TMJ arthritis can be found in other articles [11,12].

In the following sections, the methodology used to prepare the article, an overview of JIA as a medical condition together with the developments to diagnose and monitor JIA are provided.

## 2. Methodology

The methodology used to prepare this article involved thorough literature searches, using the electronic database MEDLINE (via Ovid), and the search was also cross-checked on PubMed and Google Scholar to ensure no key articles were missed. The key words used in the searches were as follows:JIA and rheumatoid arthritis,Aetiology of JIA,Clinical presentation of JIA,JIA diagnosis,Differential diagnosis of JIA,JIA management,Natural history of JIA,JIA and imaging modalities,Infrared thermal imaging and JIA,Three-dimensional imaging and JIA,JIA and accelerometry,JIA and inertia measurement unit,JIA and joint movement measurement,Artificial intelligence and JIA,Artificial neural networks and JIA,Machine learning and JIA,Fuzzy logic and JIA.

The search strategy aimed to identify and include articles that covered the diagnosis of JIA and more specifically articles that identified current and novel tools available to assist with the diagnosis of JIA. Articles were carefully screened to select those most representative of research in the field. These articles were then selected according to their relevance, publication in a reputable journal and any research focusing on only one subtype of JIA was excluded. Duplicate articles and articles not available in English were excluded, and more recently published articles were prioritized. Articles with overlapping information were reduced to the most representative to provide an accurate summary of the subject matter.

## 3. JIA and Rheumatoid Arthritis

JIA shares both clinical and pathological similarities with adult-onset rheumatoid arthritis (RA); however, they remain distinct conditions. Like JIA, RA is an inflammatory progressive disease that, untreated, can lead to joint destruction and disability [13]. Both conditions also have a complex genetic component involving human leukocyte antigen (HLA) [14]. However, RA is a much more homogenous condition than JIA, as it is classed as a single disease with different clinical manifestations as opposed to JIA, where subtypes are diagnosed as separate entities [14]. RA also tends to have worse disease outcomes compared to JIA, which has varied outcomes depending on subtype and severity of disease [14].

## 4. Aetiology of JIA

The exact pathogenesis of JIA remains unclear, though it is widely thought to be caused by an immunogenic mechanism resulting from both genetic and environmental factors [15]. One specific genetic association linked to JIA is found in the HLA region. HLA-A2 shows associations with early-onset disease, HLA-B27 is associated with enthesitis-related arthritis and other subtypes of JIA are associated with specific alleles of the HLA gene [2].

Possible environmental stimuli include infection, childhood antibiotics, maternal pregnancy, smoking, gut microbes, stress and trauma [2,15]. The genetic component of JIA is thought to be significant as monozygotic twin concordance of JIA is between 25% and 40%, showing 250–400 times increase over population prevalence, and sibling concordance is 15–30 times above population prevalence [2]. The developing opinion that JIA is a more heterogenous disease than previously thought suggests causation is likely to be multifactorial [16].

The immunogenic mechanism of JIA is believed to be mediated by chemokines that selectively attract Type 1 Helper (Th1) T-cells, resulting in the cell mediated production of proinflammatory cytokines such as interleukin-2 (IL-2), interferon-gamma and tumour necrosis factor-alpha (TNF-alpha) [2].

## 5. Clinical Presentation of JIA

Chronic inflammation of the joints presents as synovitis, causing a build-up of synovial fluid and thickening of the synovial lining. Histologically, there is hyperplasia of the lining of the synovium and infiltration of the sub-lining with various inflammatory cells. This chronic inflammatory process leads to the formation of pannus which causes cartilage and bone erosions and consequent joint damage [17]. These pathophysiological changes present clinically as a painful, red, swollen joint(s) with limited range of movement and, if the arthritis is prolonged, potential for joint deformities and growth disturbances [18].

Joint involvement in JIA usually presents as pain, swelling and stiffness of joints lasting more than 30 min. Stiffness is typically worse in the morning and relieved by movement [2]. The child’s developmental age at presentation may affect the clinical picture due to differences in communication ability; for example, using a limb in a different way, or development of a limp, may be the only sign(s) in younger children [3]. The onset of JIA often takes an insidious course which can result in a significant time period between onset of first symptom and diagnosis. There is also a risk of misdiagnosis, as JIA can closely resemble other conditions. Variation from the typical presentation of JIA can further complicate diagnosis [15].

The clinical presentation of JIA also varies depending on JIA sub-type. Children with systemic or polyarticular disease may present with more varied symptoms of fatigue, fever, weight loss or growth failure [3].

JIA can also cause a number of extra-articular manifestations, and these typically vary depending on the specific sub-type. For example, systemic arthritis may manifest with a fever and a rash as well as arthritis. Other extra-articular manifestations include dactylitis, lymphadenopathy and pericarditis. A frequently occurring extra-articular manifestation in other sub-types of JIA is chronic, anterior, non-granulomatous uveitis (iridocyclitis), which is most common in the oligoarthritis subtype (21%) but also occurs with varying incidence in other JIA subtypes including polyarticular disease. Uveitis may be present at the time of diagnosis (it can occur prior to the onset of arthritis), or it may develop during the course of the disease; therefore, patients should be screened regularly to prevent delay in diagnosis and reduce the risk of visual impairment [19]. The presence of extra-articular manifestations further increases the challenge of achieving an early accurate diagnosis.

## 6. Diagnosis of JIA

JIA is a diagnosis of exclusion. Thorough history-taking and clinical examination are imperative. The paediatric Gait Arms Legs Spine (pGALS) examination is a validated screening tool to help identify musculoskeletal abnormalities such as inflammation [17]. This screening tool is quick and easy to perform and has good sensitivity and specificity when compared with the regular clinical assessment performed by a consultant [17,20]. However, studies have shown that current examination techniques may underestimate joint inflammation, and that some asymptomatic joints show histological evidence of synovial inflammation, demonstrating the presence of subclinical inflammation [21]. The underestimation of joint inflammation by examination could be due to the subjectivity of a clinical examination, and the insidious onset of JIA that may not initially be clinically detectable. Therefore, relying on clinical examination alone may lead to delayed diagnoses and delayed or sub-optimal treatment. A clinical examination of 1667 joints identified 104 joints with inflammation, but ultrasound of the same joints identified 167 joints with inflammation, demonstrating the underestimation of inflammation by clinical examination alone [21]. A further study made comparison between physical examination and contrast-enhanced MRI and found that subclinical synovitis was present in 35.9% of cases of presumed clinically inactive JIA [22].

There is no specific diagnostic test for JIA [23]; however, blood tests can be used to exclude other diseases, assist with understanding the subtype of JIA or to help guide future treatment and management. The monitoring of non-specific inflammatory markers (erythrocyte sedimentation rate (ESR) and C-reactive protein (CRP)) can aid the diagnosis and monitoring of JIA. [17]. Certain tests can aid the specific subtype diagnosis of JIA such as anti-nuclear antibodies (ANA), rheumatoid factor (RF) and anti-cyclic citrullinated peptide antibody (Anti-CCP) [2]. These tests, however, are not specific for JIA, and levels can be raised in healthy children or by other non-JIA diseases, so their results should be interpreted with caution [17]. The use of specific biomarkers has also been explored to aid the diagnosis of JIA, particularly in systemic-onset arthritis. However, due to the complex immune reaction, many unique biomarkers have been identified, though none have been recently validated for use [24].

Imaging is regularly used as a diagnostic adjunct in JIA to support clinical evaluation [25]. Plain radiographs have previously been the traditional first-line imaging of choice. However, increasingly more varied imaging techniques are being employed to detect active inflammation such as musculoskeletal ultrasonography and contrast enhanced magnetic resonance imaging (MRI) [25]. These imaging tools contribute greater diagnostic value than plain radiograph imaging, but they are relatively expensive and remain less accessible than is needed for routine use in the outpatient clinic setting. These tools may be particularly useful in assessing joints such as the TMJ and hip, which are harder to assess clinically (as swelling and warmth are rarely appreciated) compared with joints such as the knee or ankle. Despite these tools, there is still no definitive diagnostic test for JIA; therefore, there is a pressing need to develop novel diagnostic techniques to improve the speed and accuracy of diagnosis.

## 7. Differential Diagnoses

The early diagnosis of JIA is critical to avoid permanent damage and disability [26], but this can often be challenging due to the numerous possible differential diagnoses.

An important differential diagnosis of a single swollen joint is septic arthritis; in this case, the presentation would likely include fever and more severe joint pain. This diagnosis can usually be excluded by joint aspiration with microbiological examination and specific antigen testing. Any presentation of inflammatory joint disease with symptoms atypical to JIA or with systemic symptoms should raise suspicion of malignancy. The early presentation of JIA and malignancy can be very similar; therefore, if malignancy is suspected, early investigations to exclude this differential (such as bone marrow biopsy or further imaging) are essential to rule it out [17].

Many other conditions and normal variants can mimic JIA, which adds to the diagnostic challenge when considering this diagnosis of exclusion.

## 8. Management of JIA

Once diagnosed, the early treatment of JIA is critical to optimize the potential for medically induced disease remission and to avoid further joint damage. The optimal management of JIA requires the involvement of a broad multidisciplinary team including, but not limited to, a consultant paediatric rheumatologist, specialist nurse, physiotherapist, occupational therapist, ophthalmologist, psychologist and general practitioner [23,27]. Treatment involves both pharmacological and non-pharmacological interventions, and prompt referral to a tertiary specialist service is essential to confirm diagnosis and initiate proactive treatment [23].

The pharmacological interventions for JIA begin with simple treatments and can progress to more complex therapies if patients do not respond adequately [6]. First-line treatment is with nonsteroidal anti-inflammatory drugs (NSAIDs); these are frequently used agents in the treatment of JIA [6,28]. Intra-articular corticosteroid injections are indicated in all sub-types of JIA for local inflammation relief where the disease can be localized to individual joints [6].

More intensive treatment options include systemic glucocorticoids (e.g., prednisolone), which are indicated to treat certain extra-articular manifestations [27], widespread joint involvement and active systemic-onset JIA. However, they are less commonly used now due to their extensive adverse event profile and the development of disease modifying anti-rheumatic drugs (DMARDs) and biologic drugs [6]. Both DMARDs and biologic drugs can actively slow or halt the progression of JIA and prevent the development of long-term morbidity [6].

The development of new biologic drugs over the past 20 years has significantly improved disease course and outcomes [2]. These drugs include monoclonal antibodies, receptor antagonists or soluble cytokine receptors [6] and are used to treat JIA which has not responded adequately to DMARD treatment, or where individuals are intolerant of DMARDs. The treatment of systemic-onset JIA specifically has developed to include targeted therapies for interleukin-1 and interleukin-6, as these cytokines are recognized as playing an important role in the development of inflammation [29].

Non-pharmacological management strategies are also involved in disease management; these include, but are not limited to, nutritional monitoring, increased calcium intake, physiotherapy, aquatic exercise and orthotics [23]. Exercise therapy is also helpful for children with JIA; A 2018 systematic review found that a structured, physical therapy-led exercise program focusing on strength and flexibility may have a beneficial impact on activity outcomes in JIA [30]. Alongside active treatment of JIA, ongoing monitoring for extra-articular complications is essential. Patients should be regularly screened by an ophthalmologist due to the risk of uveitis and associated eye complications [23].

## 9. Natural History of JIA

The reported prognosis of JIA is unclear, and there are varied and conflicting published data on the topic. Previous studies have shown that only 40–60% of patients achieve clinical remission or inactive disease at follow up [16]. The duration of remission achieved is also highly variable, with one study showing that only 36% of cases of clinical remission lasted two years off medication, and only 6% of cases of clinical remission lasted for five years [31]. Their study involved 437 JIA patients and examined 878 total episodes of active disease. However, it did not include analysis of how treatment type affects disease course, and this could be a topic for future research. This was also a long duration retrospective study and, as the diagnostic classification of JIA has changed over this time, the classification system used in this study may not be entirely transferable to the classification used today [31].

The diagnosed sub-type of JIA has a significant effect on predicted disease outcomes. Oligoarthritis is the most likely sub-type to achieve clinical remission, and RF-positive polyarticular JIA is the least likely to achieve remission [16,31]. Systemic-onset JIA is associated with the worst prognosis, with 30% of patients suffering from chronic polyarthritis resulting in morbidity [7]. Systemic-onset JIA has a significant non-articular disease component that has a large impact on disease morbidity.

The damage caused by JIA can affect patients into adulthood and cause chronic disability. Up to half of young adults will continue to have active disease, and up to one third will have chronic disability into adulthood [3]. The long-term damaging effects on bone and joint development include local growth disturbances at the sites of inflammation. This can result in either overgrowth or undergrowth, thereby resulting in limb length discrepancies [16]. The extra-articular uveitis manifestation of JIA can also have long-term damaging sequalae such as glaucoma, cataracts and blindness [2].

Despite the low rate of clinical remission achieved in JIA, there has been an improvement in functional disease outcomes over the past decade. The proportion of patients that develop profound functional disability ranges between 2.5 and 10% [16]. There are several predictors of poor outcome that can help identify those susceptible to serious disease. These include greater severity or extent of arthritis at onset, symmetrical disease, early wrist or hip involvement, presence of RF, persistent active disease and early radiographic changes [16]. Shorter disease duration prior to treatment and aggressive therapy have also been shown to improve disease outcomes [2].

Increasing evidence has demonstrated that early and aggressive treatment of JIA is associated with better long-term disease outcomes. This evidence offers the suggestion that there is a “window of opportunity” to begin treatment in order to achieve the best results [26,32]. This theory adds to the importance of achieving a prompt and accurate diagnosis of JIA so that early treatment can commence, and disease outcomes can be optimized.

There are also immediate effects on the general wellbeing of patients with JIA, irrespective of their disease status. These effects have negative psychosocial and physical consequences [33]. Direct physical consequences include obesity and atherosclerosis, and psychosocial impacts include chronic fatigue and pain, as well as functional impairment and decreased health-related quality of life [33]. These direct impacts of JIA can greatly affect the functioning and wellbeing of children affected with the condition. For example, a study showed that school-age children with JIA took part in fewer sports clubs and reported physical and psychological barriers to sports engagement [34].

## 10. Imaging

The serious consequences of untreated JIA highlight the importance of a prompt and accurate diagnosis. Current diagnostic methods centre around a detailed clinical history and examination supported, where appropriate, by various imaging techniques.

Plain radiographs historically have been the first-line imaging option in JIA. They are particularly useful for the exclusion of differential diagnoses involving bone such as fracture, tumour and osteomyelitis [35]. Radiographs can also demonstrate signs such as soft tissue swelling, periarticular osteopenia, epiphyseal remodelling and widening, which can all be indicators of JIA [17]. However, in the early stages of JIA, radiographs are often normal [17], and they have also been shown to be poor in identifying active synovitis [27]. Due to the importance of early diagnosis of JIA, the identification of early non-erosive joint changes has become a greater priority, for which plain radiographs are not the most suitable tool [36].

Ultrasound is a quick, cost-effective and safe imaging alternative to aid the diagnosis of JIA [17]. Ultrasound can detect synovial, cartilage and bone abnormalities earlier than conventional radiography and has demonstrated a higher sensitivity than clinical examination [37]. Ultrasound has also been shown to detect subclinical synovitis, which clinical examination alone may not detect [37], thereby enabling earlier diagnosis and earlier commencement of treatment. However, there is a lack of standard references for paediatric joint ultrasound, which limits the diagnostic value of this technique. It is also unclear whether ultrasound can differentiate between true erosions and normal surface irregularities, especially in small joints, again limiting its diagnostic utility [38]. Another limitation of ultrasound investigation is that it is time consuming and relies on operator interpretation and experience, which can be varied [39].

Magnetic resonance imaging (MRI) is a highly sensitive imaging tool for detecting joint inflammation [38]. Contrast-enhanced MRI (CE-MRI) can distinguish between clinically active and inactive JIA, which alternative investigative methods may fail to discriminate between. A study investigated MRI data sets from 146 patients with suspected or diagnosed JIA and found that MRI could differentiate clinically active and inactive disease states by identifying synovial hypertrophy [22]. The study identified that a third of the patients who were clinically presumed to have inactive JIA showed signs consistent with active synovitis on MRI. This demonstrates the utility of CE-MRI in detecting subclinical synovitis. Despite the clinical utility of MRI, its use is limited by high cost, ability to assess only a single joint or few joints per scan and the requirement for sedation in younger children [38]. There are also additional considerations for the clinician when requesting CE-MRI, such as procedural anxiety.

Magnetic resonance imaging is particularly valuable for aiding assessment of TMJ arthritis in JIA, as it can image both soft tissue and bone and allows the assessment of changes to the joint over time [40,41]. A recent study of 96 patients with JIA and 20 non-JIA controls undertook orofacial and cone-beam computed tomography systems (CBCT) examinations to identify the initial radiological signs of JIA-induced dentofacial deformity [42]. The initial radiological signs of dentofacial deformities were found to be subtle and characterized by minor mandibular asymmetry and occlusal plane steepening.

## 11. Emerging Techniques for JIA Monitoring and Diagnosis

As outlined in the previous section, imaging plays an important role in assessment of JIA. However, due to the limitations described, there remains a need to develop novel diagnostic techniques to further support the clinician’s assessment and provide objective measures of arthritis. In this section a number of these more prominently reported techniques are reviewed. These are infrared thermal imaging, three-dimensional imaging, accelerometry, artificial neural networks and fuzzy logic.

### 11.1. Infrared Thermal Imaging

Infrared radiation is part of the electromagnetic spectrum, covering a wavelength range from about 700 nanometres to 1 mm. Materials with a temperature above absolute zero (i.e., 0 Kelvin or −273.15 °C) emit infrared radiation. Infrared radiation is invisible to the human eye and can be separated into near-, mid- and far-infrared. Its applications in medical diagnosis and monitoring have been increasingly reported [43,44].

Thermal imaging has been investigated for its utility in JIA and RA. It is a harmless technique that uses infrared radiation to detect temperature changes in tissue abnormalities [44]; hence, it could be used to identify active joint inflammation (arthritis) due to an increased temperature associated with the inflammatory process. It has been investigated for use in the identification of joint inflammation in paediatric conditions in both the knee and ankle [45,46]. A study found that thermal imaging could identify significant temperature differences in the ankle joint but not in inflamed knee joints, indicating the potential of thermal imaging for detecting joint inflammation in specific joints [45]. A further study investigated 20 patients with clinically confirmed knee arthritis and demonstrated that thermal imaging detected higher temperatures in knees with active inflammation [46]. This study also demonstrated correlation between thermal and visual imaging data. There is therefore evidence that thermal imaging has the potential to be an objective tool to assist with the diagnosis and monitoring of JIA.

### 11.2. Three-Dimensional Imaging

Three-dimensional (3D) imaging is another tool that has been investigated for diagnostic use in JIA. A proof-of-concept study was conducted to determine if 3D imaging and thermal imaging could correctly quantify inflammatory changes in joint movement [47]. Both modes of imaging demonstrated quantifiable changes in joint volume and shape and could also quantify joint changes in response to therapy. Such findings suggest that these new techniques could provide objective means of detecting and monitoring arthritis. This proof-of-concept study grouped both JIA and adult arthritis as one condition, so its validity in JIA requires further investigation. Three-dimensional imaging has also been used to assess facial morphology to identify TMJ involvement in JIA. Three-dimensional imaging identified unique morphological features that indicated to the affected side/sides and the severity of TMJ involvement [48]. This pilot study indicated the use of 3D imaging as a tool to detect the signs of TMJ involvement in JIA. However, further work is required to develop this novel imaging technique into a clinically acceptable and useful tool.

### 11.3. Motion Detection Using Accelerometer and Gyroscope

An accelerometer is a sensor for accurately measuring applied acceleration [49]. Typically, an accelerometer uses three perpendicular axes, commonly referred to as x, y and z. It can be based on a number of different transducers that include piezo-electric crystals, piezo-resistive sensors and variable capacitance. Gyroscopes are also motion sensors that can be in different forms including optical, ring laser and micro-electromechanical. Micro-Electro-Mechanical-System (MEMS) gyroscopes measure rate of rotation of an object along either one, two or three axes [50]. An accelerometer and a gyroscope could be integrated into a single device called an inertia measurement unit (IMU), thus providing greater flexibility in movement measurement.

Due to their ability to measure movement, accelerometers and gyroscopes can be utilized to quantify joint movement. Potentially, these devices could quantify the extent to which joint inflammation has constrained the movement at an affected joint, by comparing movement data with normal reference values or an unaffected related joint. For example, to analyse the extent of movement at the knee joint, an accelerometer can be placed just above and below the knee, and through trigonometry and calculus, the range of movement as well as the velocity of movement, and the acceleration and deceleration can be determined [51].

Accelerometry has previously been used as a tool for the investigation of joint range of movement in gait analysis [52], post-surgery rehabilitation analysis [53] and physical activity monitoring in RA and JIA [54,55]. Earlier studies have also investigated the use of acceleration patterns to differentiate knees affected by osteoarthritis and chondromalacia as well as between rheumatoid arthritis and spondyloarthropathy [56,57]. It is hypothesized that accelerometry could also be used to objectively assess joints affected by arthritis in JIA.

### 11.4. Artificial Intelligence and Machine Learning

Artificial intelligence and machine learning encompass a number of techniques that include artificial neural networks, fuzzy logic, expert systems and genetic algorithms.

Artificial neural networks (ANN) are adaptive systems, consisting of interconnected processing elements called neurons that learn by interacting with their environment [58]. Each connection in an ANN has an associated weight. The weight values are determined when the ANN is trained. ANNs simulate the way the human brain processes and analyses information in a simplified manner. They are trained by being presented with representative examples of the relevant data. ANNs can be grouped in a number of ways, according to whether their training is supervised or unsupervised. Multilayer perceptron [59,60] and Kohonen network (self-organizing map) [61] are examples of supervised and unsupervised learning ANN, respectively.

Fuzzy logic is a generalization of classical logic that attempts to perform reasoning by modelling human ways of thinking or reasoning [62]. Unlike crisp sets that require a measurement to belong to a specific category (set), in fuzzy logic, a measurement can belong to several sets with different degrees of memberships. Degree of membership has a continuous range between 0 and 1, with 0 indicating not a member and 1 a full membership.

Artificial intelligence and machine learning techniques have the ability to filter an abundance of information and identify clinically relevant details. This could have implications for their use in diagnosis, management, monitoring and disease risk estimation of a multitude of disorders, including those seen in rheumatology [63].

Possible applications of machine learning in rheumatology include electronic diagnosis systems, automatic electronic medical records filtering [64], learnt disease prediction models [65], interpretation of genetic markers and image recognition [66,67]. It has been theorized that in the future, machine learning could be utilized to assist rheumatologists in predicting the course of disease and identifying disease factors, and through reinforcement learning, it may be able to make treatment propositions [67]. Through these methods, AI could possibly be used as a tool to support rheumatologists in providing a timely and accurate diagnosis to help create effective management strategies, and to monitor for flares of inflammation. All these elements are important in achieving optimum outcomes in JIA. There has been little research on the use of these techniques specifically applied to JIA; however, they have been used more broadly in the medical field and in rheumatoid arthritis [68].

AI has been investigated for its application specifically analysing imaging in rheumatology for many years; however, most methods have yet to enter clinical practice [68]. A certain subtype of AI, called deep learning, has shown potential to interpret images beyond the human-level of accuracy [68]. Potentially this could be applied to the interpretation of imaging in JIA in order to improve diagnostic accuracy. Deep learning is a sub-field of machine learning that uses large neural networks to mimic human decision making [67]. Deep learning has been applied in RA to aid the interpretation of different imaging modalities. For example, it has been shown capable of detecting bone erosions from MRI scans and identifying patients with RA from healthy subjects from an X-ray of the hand [69].

AI has also been shown to measure the extent of synovitis and allocate a score from Doppler ultrasound images [66]. However, it is unlikely that AI could replace radiologist interpretation in the near future, as the generalization of computerized image interpretation would be extremely difficult in such a broad field. It is more likely that AI could work with human intelligence as a hybrid solution to improve image assessment [68]. In order to establish such systems, large amounts of data are needed to train the system, and if these can only be obtained through human interpretation then there is less to gain. Further research into methods such as neural networks and alternative reference outcome measures is required to ascertain if this technology could be employed in routine clinical care within rheumatology [68].

Although more limited, AI has been explored directly within JIA. It has been used to create automated patient education dialogues for families of children with JIA. A patient-led dialogue was created using AI techniques and was evaluated by six rheumatology specialists who found the dialogue was able to provide accurate, relevant and mostly complete information [70]. This study demonstrates the potential broader use of AI within rheumatology to help address some of the gaps in patient education experienced by families affected by JIA.

Fuzzy logic has gained attention in medical diagnosis and monitoring due to its ability to model and accurately represent uncertainty. Although we did not find a publication related to its use specifically in JIA, a study used fuzzy logic to develop a relationship map between rheumatic-musculoskeletal symptoms to risk factors [71]. A fuzzy logic-based system has been devised to assist with diagnosis of arthritis [72]. Fuzzy logic models that could provide diagnostic confidence at different levels for RA were reported [73].

The use of AI in rheumatology, and JIA more specifically, is relatively unexplored. Artificial intelligence-based systems could potentially perform complex and highly specific clinical tasks; however, further work is required to develop these systems for routine clinical use. The importance of an early and accurate diagnosis in JIA makes AI an attractive potential avenue for future research.

## 12. Overall Discussion

With advancements in care, medicines, research and technology there is an ever-growing desire to achieve earlier diagnosis for children and young people who develop JIA. JIA is primarily a diagnosis of exclusion and relies upon the subjective history-taking and clinical examination; therefore, the clinician often relies upon the use of diagnostic imaging to support decision-making. The traditional options of X-ray, musculoskeletal ultrasound and CE-MRI scanning, have benefits and limitations and, for X-ray, potential exposure risks when used periodically. There is the need for innovative, acceptable and accessible tools to help the clinician confirm or refute a suspicion of subtle joint inflammation early in the course of JIA.

New innovative techniques can complement traditional methods of JIA diagnosis and monitoring. The application of accelerometry allows joint movements to be quantified and thus provide an objective measure of joint restriction. For example, when applied to the knee joint, the method provides information such as the extent of movement (angle) and movement features through angular velocity and acceleration measurements. Accelerometry is cost effective and easy to apply and can be performed at the patient’s home. Machine learning and AI methods such as artificial neural networks (ANNs), fuzzy logic and expert systems allow the available patient data to be interpreted and analysed quickly and accurately, thereby aiding clinicians in making diagnostic decisions in a timelier manner. Fuzzy logic allows imprecise information to be scientifically analysed and relevant inferencing to be carried out. Fuzzy logic could be useful in JIA monitoring and diagnosis, as clinical observations are often imprecise. Extensive data are collected in the medical field, and often the full patient benefit from this data is not realized, partly because busy clinicians do not have sufficient time to analyse all the available information. Data can have great value in improving patient care when correctly processed and analysed. One of the main contributions of AI techniques is making better use of these data for the benefit of patient care.

This review has demonstrated early signs within research that engineering technologies including AI can be adapted to support the diagnostic and monitoring needs of children and young people with JIA. In our experience of conducting research within the area of thermal imaging, for example, children and young people universally report their preference of this novel technology to traditional monitoring methods, which can sometimes generate anxiety [46].

## 13. Conclusions

Early diagnosis and treatment is increasingly recognized as being essential to improve disease outcomes for children and young people with JIA. Development of novel applications of technology offers potential to help clinicians in making early diagnoses and target available treatments earlier in the course of this disease. Within engineering technology research, newer monitoring and diagnostic techniques have been shown to potentially assist with diagnosing and managing arthritis. Techniques of particular current interest are accelerometry to analyse joint movement and restriction, three-dimensional imaging to accurately visualize joints, infrared thermal imaging to precisely quantify skin surface temperature and artificial intelligence to assist clinical precision and decision making. Within AI, artificial neural networks, deep learning neural networks and fuzzy logic have gained particular attention recently.

This review has presented evidence on the current status of JIA diagnosis and management and provided a summary of current novel tools to assist with the clinical management of JIA. Further work is needed to develop these promising technologies into clinically useful and acceptable tools to help achieve earlier diagnosis of JIA, and thereby achieve the better outcomes which existing evidence suggests are possible.

## Data Availability

No data collection was involved in this study.

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
