# Peer review of "Juvenile Idiopathic Arthritis: A Review of Novel Diagnostic and Monitoring Technologies"

_healthcare, 2021, doi:10.3390/healthcare9121683_

Round 1

Reviewer 1 Report

This article is a complete review of the  recent novel tools and techniques which have potential value in assessment and monitoring of JIA is very interesting. 

I found interesting the part regarding the emerging tools and techniques because I think that some of them (AI and Machine learning) will be very useful in assessment and monitoring of JIA. As I previously wrote I think this part is complete. Advantages and disadvantages of the new tools are described, also in term of the future work that should be done to make them clinically acceptable for the assessment of JIA patients.

As I have already wrote I would add a couple of things regarding the TMJ, because it is often cited as the forgotten joint. The assessment of the TMJ involvement is slightly different compared with the other joints of the body and so I think that it should be discussed in the introduction part

The article is clear and well structured. In the imaging section I would add:

- some sentences regarding the use of CBCT, that is s tool of early diagnosis of TMJ involvement by means of morphometric measures. (Stoustrup P, Traberg MS, Matzen LH, Glerup M, Küseler A, Herlin T, Pedersen TK. Initial radiological signs of dentofacial deformity in juvenile idiopathic arthritis. Sci Rep. 2021 Jun 23;11(1):13142).

Author Response

Dear Honorary Editors, Respected Reviewers

Thank you very much for so kindly reviewing our paper and making very valuable constructive comments. We have considered the comments and have implemented them carefully. The changes made are summarised in the attached table and highlighted on the paper. The yellow highlighted areas on the paper are related to the reviewer 1 comments and green areas are related to reviewer 2 comments. The corrected typos are not highlighted on the paper. The paper has become much more enhanced following the revisions.

We are very grateful for the help and support provided and hope our revisions meet your expectations.

Best wishes

Professor R Saatchi

Reviewer 2 Report

The authors have to address comments carefully to ensure that all problems affecting the plainness of the review are removed.

  • The methodology of this review is unclear. The authors should clarify the methods for this review.
  • The discussion is very short and requires more detail.
  • English Writing: This paper requires moderate proofreading of the entire of the paper to remove all the problems related to typos, spelling, and grammar mistakes. The authors should check the paper completely to verify the integrity and accuracy of the English writing.
  • List of references: The number of references is sufficient for this study, but some references are not very relevant to the topic of the research. The references should follow Healthcare-MDPI Journal style. Some search names in the references list begin with an uppercase letter in each word such as [1] …etc., and other words begin with a lowercase letter, such as [2], [3] …etc. Authors should standardize the writing style of research names. References are not arranged in the text correctly, for example, reference [4] comes before reference [3]. This paper requires double check of the reference list.

Author Response

(The authors gave the same response as above.)

Round 2

Reviewer 2 Report

The authors responded to most of our comments, however, there is one comment that is still some minor mistakes in the English language.

  • English Writing: This paper requires minor proofreading of the entire of the paper to remove all the problems related to typos, spelling, and grammar mistakes. The authors should check the paper completely to verify the integrity and accuracy of the English writing.

Thanks